# Implications of Spectral Interlacing for Quantum Graphs

**DOI:** 10.3390/e25010109

**Published:** 2023-01-04

**Authors:** Junjie Lu, Tobias Hofmann, Ulrich Kuhl, Hans-Jürgen Stöckmann

**Affiliations:** 1Institut de Physique de Nice, CNRS, Université Côte d’Azur, 06108 Nice, France; 2Fachbereich Physik, Philipps-Universität Marburg, 35032 Marburg, Germany

**Keywords:** quantum graphs, interlacing theorem, random matrix theory

## Abstract

Quantum graphs are ideally suited to studying the spectral statistics of chaotic systems. Depending on the boundary conditions at the vertices, there are Neumann and Dirichlet graphs. The latter ones correspond to totally disassembled graphs with a spectrum being the superposition of the spectra of the individual bonds. According to the interlacing theorem, Neumann and Dirichlet eigenvalues on average alternate as a function of the wave number, with the consequence that the Neumann spectral statistics deviate from random matrix predictions. There is, e.g., a strict upper bound for the spacing of neighboring Neumann eigenvalues given by the number of bonds (in units of the mean level spacing). Here, we present analytic expressions for level spacing distribution and number variance for ensemble averaged spectra of Dirichlet graphs in dependence of the bond number, and compare them with numerical results. For a number of small Neumann graphs, numerical results for the same quantities are shown, and their deviations from random matrix predictions are discussed.

## 1. Motivation

Quantum graphs are composed of bonds which are connected with each other at vertices. Along the bonds wave propagation is governed by the Schrödinger equation without potential and boundary conditions depending on the details of the coupling at the vertices. Quantum graphs were first introduced by Pauling [1] in the context of free electron models of organic molecules. Later, they were studied intensely in physics [2] and mathematics [3], and experimentally implemented in correspondingly-shaped microwave networks [4]. They are conceptually simple, but still complex, and there is a straightforward symbolic alphabet to classify the periodic orbits. Casati and coworkers [5] suggested that the universal features of the spectra of chaotic systems might be described by random matrix theory (RMT), which later was expressed by Bohigas, Giannoni, and Schmit [6] in the form of a conjecture. Using supersymmetry techniques, Gnutzmann and Altland [7] proved the conjecture for the two-point correlation function for fully connected graphs with incommensurate bond lengths. Their result was generalized to all correlation functions by Pluhař and Weidenmüller [8]. Just as for billiard systems [9], there is a one-to-one correspondence between a quantum graph and the corresponding microwave networks, which has been used, in particular, by Sirko and coworkers in numerous experiments to study spectral and scattering properties of microwave graphs (see e.g., Ref. [4]).

In a recent microwave experiment in tetrahedral graphs [10], however, we noticed that one important aspect is missing in the above scenario. It is hidden in the structure of the equation system determining the graph spectrum. Using energy and current conservation (the Kirchhoff rules in experimental networks), one arrives at a secular equation [2]
(1)$$\sum_{m=1}^{V}h_{nm}\varphi_{m}=0\;,$$
where the sum runs over all vertices *V*, and $\varphi_{m}$ is the potential at vertex *m*. In the experiment, the bonds are connected by ordinary T junctions corresponding to Neumann boundary conditions. For this situation, the elements of the secular matrix *h* are given by
(2)$$h_{nm}=-\delta_{nm}\sum_{m^{\prime }}\cot kl_{nm^{\prime }}+\frac{1}{\sin kl_{nm}}\;,$$
where the $l_{nm}$ are the lengths of the bonds connecting vertices *n* and *m*, and *k* is the wave number. For the homogeneous equation system (Equation 1) to have non-trivial solutions, the determinant of h(k) has to vanish,
(3)$$\left|h(k)\right|=0\;.$$

The roots $k_{n}$ of the equation generate the spectrum of the graph. It will be called “Neumann” spectrum in the following since the T junctions at all vertices obey Neumann boundary conditions. On the other hand, $h_{nm}$ becomes singular, whenever $kl_{nm}$ is an integer multiple of π. This situation corresponds to a totally disassembled graph with a spectrum being the sum of the spectra of all individual bonds with Dirichlet boundary conditions at both ends, thus the vertices have no influence any longer. This “Dirichlet” spectrum hence appears via the *poles* of |h(k)|, whereas the Neumann spectrum is given by the *zeros* of |h(k)|. In the following, all lengths will be assumed to be incommensurable to avoid degeneracies of the Dirichlet spectrum.

This “spectral duality”, as we termed it in our previous publication [10], has important consequences for the spectral statistics. The cause is the interlacing theorem (see, e.g., Chapter 3.11 of Ref. [3]): *If the boundary conditions at one vertex of a graph are changed from Neumann to Dirichlet, or somewhere in between, the eigenvalues of the original and the new graph appear strictly alternating.*

To move from the Neumann to the Dirichlet spectrum for a complete graph, the boundary conditions have to be changed one after the other at *all* vertices, not just at one of them. Now, there is no longer a strict alternation in the sequence of the respective eigenvalues, but still a strong correlation remains—the maximum number of Neumann eigenvalues confined between two successive Dirichlet ones is given by the number of vertices *V*, and vice versa.

The mean density of states for a graph of a total length of $l_{\mathrm{tot}}$ is given by
(4)$$\bar{\rho }\left(k\right)=\frac{l_{\mathrm{tot}}}{\pi }\;.$$

For a graph with *B* bonds and a given total length, the maximum level spacing is found for the limiting case where all bonds are equal. For this case, the Dirichlet spectrum is *B*-fold degenerate, and the maximum distance between neighboring eigenvalues, in units of the mean level spacing, is just $s_{\max}=B$. Due to the interlacing theorem, the same must be true for the Neumann resonances. There is hence a cut-off in the level spacing distribution p(s) at $s_{\max}=B$, at the latest, both for the Dirichlet and the Neumann spectrum. Consequences of spectral interlacing for the number variance have been discussed already in our previous paper [10].

Thus, there are clear deviations from the RMT expectation for small graphs. This is not in contradiction with the proofs mentioned in the beginning that the spectra of graphs with irrational length ratio *do* exhibit RMT behavior, since these proofs work in the limit of infinitely large graphs only. From the practical point of view this is of little help since numerical, as well as experimental, studies are necessarily restricted to comparatively small graphs.

Therefore, an understanding of the impact of Dirichlet–Neumann interlacing is mandatory for the correct interpretation of the spectral statistics in small graphs. Since, in the moment, a good idea to approach Neumann spectral statistics is still missing, we start with a more modest task—the interpretation of Dirichlet spectral statistics. Analytic results are given for level spacing distribution and number variance for a random superposition of lattice fence spectra and compared with numerical results. For the Neumann spectra, we restrict ourselves to an illustration of the fingerprints of spectral interlacing in level spacing distribution and number variance, but have to leave the theoretical interpretation to future papers. We do not discuss experimental results from microwave graphs in the present paper. This remark may be necessary since probably this is exactly what readers do expect from our group.

## 2. Dirichlet Graphs

For Dirichlet graphs, there are Dirichlet boundary conditions at each end of all bonds, thus the bonds are not coupled at the vertices and the spectrum corresponds to a superposition of *B* separated bonds. Here, we present analytical and numerical results of the spectral statistics for ensemble-averaged Dirichlet graphs. Following the usual practice, the mean density of states $\bar{\rho }\left(k\right)=l_{\mathrm{tot}}/\pi$ was kept constant and normalized to one, meaning a total length of $l_{\mathrm{tot}}=\pi$ for all graphs entering the average. For the numerics, the lengths had been created by generating B−1 random numbers $r_{n}$ between 0 and π, and by taking the appearing *B* segments as lengths $l_{n}$. The procedure yields $p_{B}\left(l_{1},\cdots ,l_{B}\right)=\frac{1}{\pi^{B-1}}\delta \left(\sum l_{n}-\pi \right)$ for the joint length probability. The $l_{n}$ are hence uniformly distributed on the interval 0 to π with the constraint $\sum l_{n}=\pi$. Integrating out all $l_{n}$ but one obtains the distribution
(5)$$p_{B}\left(l\right)=\frac{B-1}{\pi^{B-1}}{\left(\pi -l\right)}^{B-2}$$
for the remaining *l*, being constant only for B=2. The derivation and a plot of the length distributions can be found in Appendix A.

Alternatively, one could think of taking *B* lengths $l_{n}^{\prime }$ from an interval between 0 and 1, and afterward, normalizing each length via $l_{n}=\pi l_{n}^{\prime }/\sum_{n=1}^{B}l_{n}^{\prime }$ to a mean density of one, i.e., $\sum l_{n}=\pi$. The resulting joint length probability is non-uniform, in contrast to the one above. For the sake of conciseness, we shall refer in the following to the two respective ensembles as the uniform and the non-uniform one. The non-uniform approach would be more in the spirit of the usual unfolding technique used in quantum chaos to make spectra taken from different systems comparable. For the non-uniform ensemble again, numerical length distributions are presented in Appendix A. Since it would be hard to obtain analytical results for the non-uniform ensemble, all analytics and numerics, if not explicitly stated differently, are for the uniform one.

In the next two subsections, theoretical expressions for nearest neighbor spacing distribution p(s) and number variance $\Sigma^{2}$ are given and compared with numerical data.

### 2.1. Nearest Neighbor Spacing Distribution for Dirichlet Graphs

To calculate the distribution of nearest neighbors spacings p(s) for the Dirichlet spectrum of a graph, we apply a strategy that had been used already by Berry and Robnik [11] to calculate p(s) for an uncorrelated superposition of two spectra, one associated with the chaotic part, the other with the regular part of a mixed phase-space system. A key element in the calculation is the gap probability e(s) describing the probability for a spectral range of length *s* to be empty of eigenvalues. The gap probability is related to the level spacing distribution via
(6)$$p\left(s\right)=e^{\prime \prime }\left(s\right)\;,$$
where a mean level spacing of one has been assumed. Expression (Equation 6) is well-known to those working in the field, but for readers not familiar with the subject, a didactic derivation is given in Appendix B. For a picket fence spectrum with a mean level spacing of Δs=1, the gap probability is given by
(7)$$e\left(s\right)=\left\{\begin{array}{cc} 1-s, & \mathrm{if}\;0\leq s\leq 1\;,\\ 0, & s>1\;. \end{array}\right.$$

e(s) is, in contrast to p(s), multiplicative for superimposed uncorrelated spectra,
(8)$$e\left(s\right)=\prod_{n}e_{n}\left(s\right)\;,$$
whence follows for the Dirichlet spectrum of a graph with *B* bonds of lengths $l_{n}$, n=1,⋯,B
(9)$$e_{B}\left(k\right)=\prod_{n=1}^{B}e\left(\frac{l_{n}}{\pi }k\right)\;.$$

From Equation (Equation 9) for $e_{B}\left(k\right)$, the Dirichlet level spacing distribution can now be obtained by taking the second derivative, see Equation (Equation 6). The Dirichlet level spacing distribution has already been calculated by Barra and Gaspard [12], who did not follow, however, the approach of Berry and Robnik [11]. Their derivation therefore is much less concise and considerably longer then the present one, see the appendix of [12].

Expression (Equation 9) has to be averaged over all different realizations of $l_{n}$ with the constraint that the total length $l_{\mathrm{tot}}$ is constant
(10)$$\sum_{n=1}^{B}l_{n}=l_{\mathrm{tot}}\;.$$
with the substitution $s_{n}=\frac{l_{n}}{\pi }k$, the constraint becomes
(11)$$s=\sum_{n=1}^{B}s_{n}=\frac{l_{\mathrm{tot}}}{\pi }k\;.$$

Thus, *s* is the wave number in units of the mean level spacing $\pi /l_{\mathrm{tot}}$. In the following we shall use the letter *s* exclusively for spectra with a mean level spacing of Δs=1. Now the average can written as
(12)$$\begin{array}{cc} \langle e_{B}\left(s\right)\rangle = & \frac{(B-1)!}{s^{B-1}}\int_{0}^{s}ds_{1}e\left(s_{1}\right)\int_{0}^{s-s_{1}}ds_{2}e\left(s_{2}\right)\;\cdots \;\int_{0}^{s-s_{1}-\cdots -s_{B-2}}ds_{B-1}e\left(s_{B-1}\right)\\ \; & \cdot e(s-s_{B-1}-\cdots -s_{1})\\ = & \frac{(B-1)!}{s^{B-1}}w_{B}\left(s\right)\;, \end{array}$$
where $w_{B}\left(s\right)$ is given by
(13)$$\begin{array}{cc} w_{B}\left(s\right) & =\int_{0}^{s}ds_{1}e\left(s_{1}\right)\int_{0}^{s-s_{1}}ds_{2}e\left(s_{2}\right)\;\cdots \\ \; & =\int_{0}^{s}ds_{1}e\left(s_{1}\right)w_{B-1}\left(s-s_{1}\right)\;,\mathrm{with}\;w_{1}\left(s\right)=e\left(s\right)\;. \end{array}$$

The factorial in Equation (Equation 12) reflects the number of possible *l* sequences to do the average. Equation (Equation 13) can be used to calculate $w_{B}\left(s\right)$ iteratively. For B=2, e.g., one obtains
(14)$$w_{2}\left(s\right)=\left\{\begin{array}{cc} \int_{0}^{s}ds_{1}e\left(s_{1}\right)e\left(s-s_{1}\right)=s-s^{2}+\frac{1}{6}s^{3}\;, & s<1\;,\\ \int_{s-1}^{1}ds_{1}e\left(s_{1}\right)e\left(s-s_{1}\right)=\frac{4}{3}-2s+s^{2}-\frac{1}{6}s^{3}\;, & 1<s<2\;,\\ 0\;, & s>2\;, \end{array}\right.$$
where the limits of integration in the different *s* windows take care of the cut-off of e(s). With help of Equations (Equation 12) and (Equation 6), we now obtain for the level spacing distribution
(15)$$p_{2}\left(s\right)=\left\{\begin{array}{cc} \frac{1}{3}\;, & 0<s<1\;,\\ \frac{8-x^{3}}{3x^{3}}\;, & 1<s<2\;,\\ 0\;, & s>2\;. \end{array}\right.$$

In this way, the $p_{B}\left(s\right)$ may be obtained iteratively, resulting in formulas with a complexity increasing step by step.

A more direct approach takes advantage of the fact that the integral in Equation (Equation 13) is nothing but a convolution. In such a situation, Laplace transform techniques are the method of choice. Applying a Laplace transform to Equation (Equation 13), the convolution theorem yields
(16)$${\hat{w}}_{B}\left(\lambda \right)=\hat{e}\left(\lambda \right){\hat{w}}_{B-1}\left(\lambda \right)\;,$$
where
(17)$${\hat{w}}_{B}\left(\lambda \right)=L\left[w_{B}\left(s\right)\right]=\int_{0}^{\infty }w_{B}\left(s\right)e^{-\lambda s}ds$$
and
(18)$$\hat{e}\left(\lambda \right)=L\left[e\left(s\right)\right]=\int_{0}^{\infty }e\left(x\right)e^{-\lambda x}dx=\frac{1}{\lambda^{2}}\left[e^{-\lambda }-1+\lambda \right]$$
are the Laplace transforms of w(s), and e(s), respectively. Iterating Equation (Equation 16), one gets
(19)$${\hat{w}}_{B}\left(\lambda \right)={\left[\hat{e}\left(\lambda \right)\right]}^{B}\;,$$
whence $w_{B}\left(s\right)$ is obtained via an inverse Laplace transform
(20)$$w_{B}\left(x\right)=L^{-1}\left({\hat{w}}_{B}\left(\lambda \right)\right)\;.$$

The inverse Laplace transform can be done with the result
(21)$$w_{B}\left(s\right)=\left\{\begin{array}{cc} 0\;, & s>B\;,\\ \sum_{m=0}^{\left\lfloor s\right\rfloor }\sum_{l=m}^{B}c_{ml}^{B}{\left(-1\right)}^{l-m}{\left(s-m\right)}^{l+B-1}\;, & s<B\;, \end{array}\right.$$
where ⌊s⌋ denotes the largest integer ≤s and
(22)$$c_{ml}^{B}=\frac{B!}{m!(l-m)!(B-l)!(B+l-1)!}\;.$$

Further details can be found in Appendix C.

To verify our results, we compare the analytical results with numerical simulations. In Figure 1, the histograms for B=2 to 6 and 100 are shown together with the corresponding theoretical predictions in a linear, and in Figure 2 in a logarithmic scale. All distributions show the expected cut-off at $s_{\max}=B$. There is a perfect agreement between numerics and theory. Note the discontinuity for B=2 at s=1, which for larger *B* is smoothed out and vanishes for B→∞, where an an exponential decay is expected, corresponding to a Poisson distribution. This can be seen in Figure 1f and Figure 2f, showing the results for B=100. There are still deviations from the exponential behavior as can be seen in the inset of Figure 2f, showing the same results over a larger *s* range. Still the analytic solution matches better. In Appendix D, numerical findings are presented for the non-uniform ensemble.

### 2.2. Number Variance for Dirichlet Graphs

The number variance, defined as
(23)$$\Sigma^{2}\left(s\right)=\langle n^{2}\rangle -{\left(\langle n\rangle \right)}^{2}\;,$$
where *n* is the number of eigenvalues in an interval of length *s*, yields for the lattice fence spectrum of a single bond of length *l*
(24)$$\Sigma^{2}\left(s\right)=\left\{s\right\}\left[1-\left\{s\right\}\right]\;,\mathrm{with}\;s=\frac{kl}{\pi }\mathrm{and}\left\{s\right\}=s-\left\lfloor s\right\rfloor \;.$$

It is convenient to express $\Sigma^{2}\left(s\right)$ in terms of its Fourier transform,
(25)$$\Sigma^{2}\left(s\right)=\frac{1}{6}-\frac{1}{\pi^{2}}\sum_{m=1}^{\infty }\frac{\cos(2\pi ms)}{m^{2}}\;.$$

For *B* bonds with independent bond lengths $l_{n}$, the spectrum is just the superposition of the *B* spectra with bond lengths $l_{1},l_{2},\cdots ,l_{B}$. The number variation $\Sigma^{2}\left(s\right)$ is additive for uncorrelated spectra leading to
(26)$${\langle \Sigma^{2}\left(s\right)\rangle }_{l}=B\left\{\frac{1}{6}-\frac{1}{\pi^{2}}\sum_{m=1}^{\infty }\frac{{\langle \cos\left(2\pi ms_{k}\right)\rangle }_{l}}{m^{2}}\right\}$$
for the ensemble averaged number variance, where $s_{k}=\frac{kl_{k}}{\pi }$, and ${\langle \cdots \rangle }_{l}$ means the average over all $l_{n}$ with the constraint $\sum l_{k}=l_{\mathrm{tot}}$, i.e.,
(27)$${\langle \cos\left(\alpha l_{k}\right)\rangle }_{l}=\int_{0}^{\pi }dl\;p_{B}\left(l\right)\cos\left(\alpha l\right)\;,$$
with α=2mk and $p_{B}\left(l\right)$ given by Equation (Equation 5).

In Figure 3, the ensemble averaged number variance for Dirichlet graphs are shown for a number of different bonds. For a single bond (B=1), there is just one lattice fence spectrum with a spacing of one. Hence, one observes a periodic modulation with an average value of 1/6, as described by Equation (Equation 25). With increasing *B*, these oscillations are damped out more and more, until $\Sigma^{2}\left(s\right)$ eventually approaches the linear increase expected for a Poissonian ensemble. A good agreement between the simulations and the analytical predictions given by Equation (Equation 27) is found.

## 3. Neumann Graphs

Here, we present the results of the Neumann graphs shown in Table 1. We restricted ourselves to graphs where bonds are connected at least to two other bonds and where there are no disconnected parts. In addition the verticity $V_{B}$, the number of bonds that connect at a vertex, has been assumed to be the same for all vertices. The smallest graph of interest is the tetrahedron which has been used repeatedly for RMT studies in numerics [13], as well as in experiments [4,10].

### 3.1. Nearest Neighbor Spacing Distribution for Neumann Graphs

Figure 4 shows level spacing distributions for the graphs presented in Table 1. In addition, the exact RMT nearest neighbor distribution is shown [14,15]. Whereas the linear plot suggests a reasonable agreement with the RMT prediction, in the logarithmic plot for all four graphs, a suppression for large values becomes obvious, with the strongest suppression for the tetrahedron having the smallest number of bonds. This is in accordance with the expectation; due to the interlacing theorem, the largest possible distance is given by the number of bonds, B=6 for the tetrahedron, and 10 or 12 for the other graphs. The decay, however, does not increase monotonously with *B*—it is faster for the hexahedron than for the octahedron, though the number of bonds is the same, and the decay for the fully connected five-vertices graph is as fast as for the octahedron, though the number of bonds is not the same.

Similar deviations of p(s) from RMT have also been observed by Barra and Gaspard [12] but not discussed in detail.

To quantify these findings, we determined $s_{c}$, the value where p(s) drops below $10^{-5}$, i.e., $p\left(s_{c}\right)=10^{-5}$, close to the limit of our statistical precision. In Figure 4, this *s* value reflects the point where p(s) crosses the abscissa. The extracted values are presented in Table 1. Regrettably, it is impossible to fix the cut-off point by the numerics, which should be, e.g., $s_{\max}=6$ for the tetrahedron. This would need more than $10^{11}$ spacings, by far beyond our computer resources.

### 3.2. Number Variance for Neumann Graphs

In Figure 5, the ensemble-averaged number variances for the graphs shown in Table 1 are plotted, exhibiting a saturation at about s=2 in contrast to the behavior predicted by RMT. Similar deviations from RMT predictions for the number variance in graphs have been reported in Refs. [16,17], and for the spectral rigidity in Ref. [4]. For non-experts, we mention that the spectral rigidity may be looked upon as a smoothed version of the number variance (the exact definition is technical and not of relevance here [14]). Already in 1985, Casati and coworkers [18] discovered a saturation of the spectral rigidity in the spectra of rectangular billiards, which could be traced back by Berry to the influence of the shortest periodic orbit [19]. In the present case this can not be the explanation. From the shortest periodic orbit, there should be a saturation of $\Sigma^{2}\left(s\right)$ at about $s_{\mathrm{sat}}=\pi /l_{\min}=l_{\mathrm{tot}}/l_{\min}$. Since necessarily $l_{\min}\leq l_{\mathrm{tot}}/B$, periodic orbit theory predicts a saturation of the number variance not until $s_{\mathrm{sat}}\geq B$, whereas actually for all graphs the saturation is observed much earlier at about s=2. In the lower part of Table 1, the saturation values are given, as well as $s_{m}$ corresponding to the *s* value where $\Sigma^{2}\left(s\right)$ is maximal. $s_{m}$ is a convenient tool to quantify the point of cross-over from a linear increase of $\Sigma^{2}\left(s\right)$ for small *s* values to a saturation for s→∞. There is a clear correlation between $s_{c}$, $s_{m}$, and $\Sigma_{\mathrm{sat}}^{2}$, collected in the lower part of Table 1. From the interlacing theorem one would expect a correlation of these quantities with the bond number, which is observed for the first three graphs presented in the table, the tetrahedron, the totally connected graphs with five vertices, and the octahedron, but the hexahedron does not fit into the sequence. In fact, there is a stronger correlation with the vertex valency, the number of bonds meeting at a vertex. Obviously, the interlacing theorem alone is not sufficient to describe all these features.

A quantitative explanation, in particular, of the saturation values, has to be postponed to further studies, but there is a qualitative explanation. Semiclassical theory relates RMT to periodic orbits [20]. Essential ingredients are correlations between orbits and their time-reversed partners, in case there is time-reversal symmetry [19], and between various types of self-intersecting orbits with their non-intersecting partners [21,22]. Apart from this, all orbit lengths are assumed as uncorrelated. This assumption is severely violated in graphs, where all orbits are composed from a finite number of elements.

## 4. Conclusions

The implications of spectral interlacing in quantum graphs have been discussed. For Dirichlet graphs, explicit analytic expressions have been obtained for level spacing distribution and number variance, and compared with numerical results. For Neumann graphs, numerical results for the same quantities have been presented, showing clear deviations from RMT predictions due to spectral interlacing. For Neumann graphs, an analytic description of these features is still missing and has to be left to future work.

## Figures and Tables

**Figure 1 entropy-25-00109-f001:**
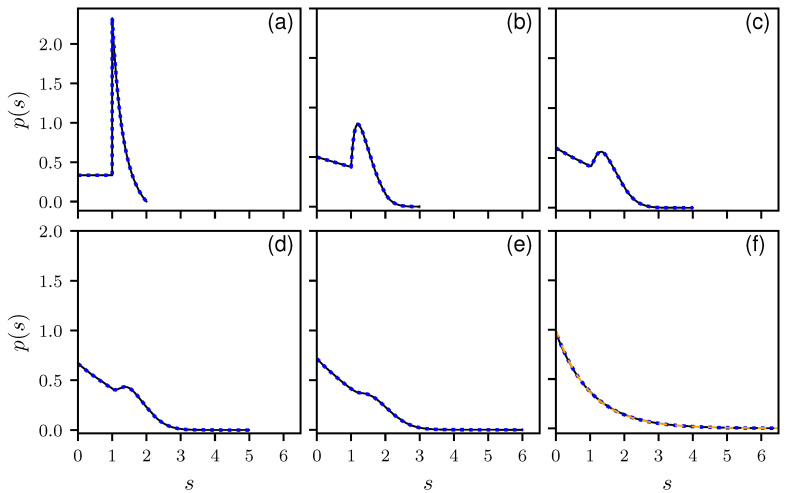
The distribution of nearest neighbor spacings of the Dirichlet graphs for different numbers of bonds B=2 (**a**), 3 (**b**), 4 (**c**), 5 (**d**), 6 (**e**), 100 (**f**) in linear scale. The solid lines correspond to numerical simulations taking into account $10^{9}$ realization, each of them containing about 900 spacings. The blue dotted lines corresponds to the theoretical prediction, Equations (Equation 6) and (Equation 12). In (**f**), the orange dashed line corresponds to a Poisson distribution, i.e., an exponential.

**Figure 2 entropy-25-00109-f002:**
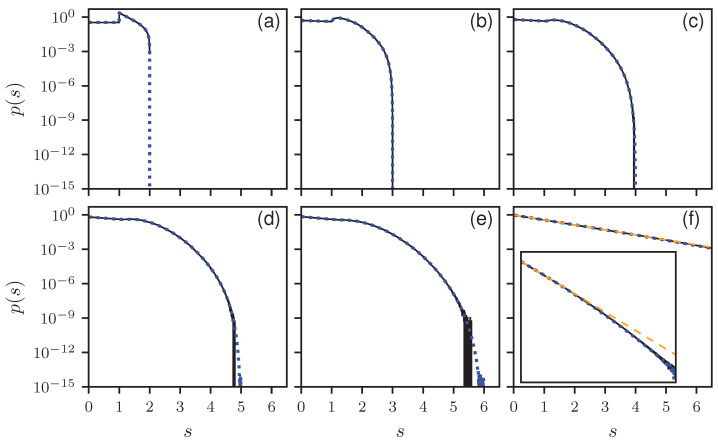
The same as Figure 1 but in logarithm scale. In the inset in (**f**), the abscissa ranges from 0 to 16, and the ordinate from $10^{-9}$ to 5.

**Figure 3 entropy-25-00109-f003:**
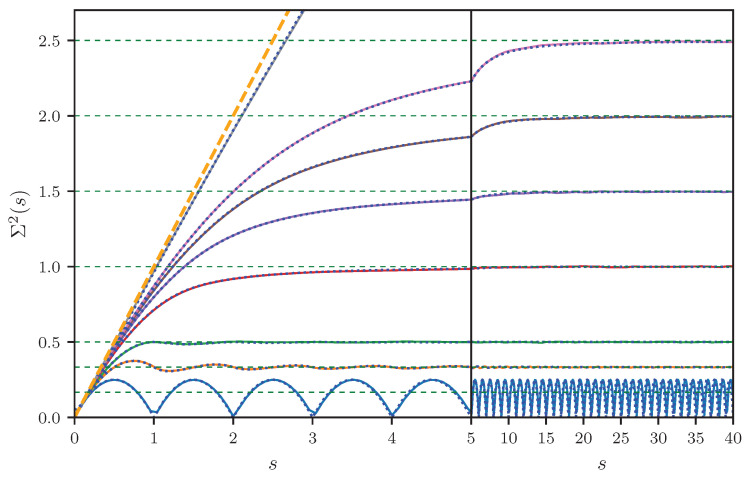
Ensemble averaged number variance $\Sigma^{2}\left(s\right)$ for the Dirichlet graphs with B=1,2,3,6,9,12,15,100 bonds. The solid lines correspond to the numerical simulations and the blue dotted lines to the analytical result given by Equation (Equation 26). The horizontal green dashed lines mark the limit $\Sigma^{2}\left(s\right)\to B/6$ for s→∞. The straight orange dashed line represents the number variance for integrable systems given by $\Sigma^{2}\left(s\right)=s$. Note the change of the abscissa scale at s=5.

**Figure 4 entropy-25-00109-f004:**
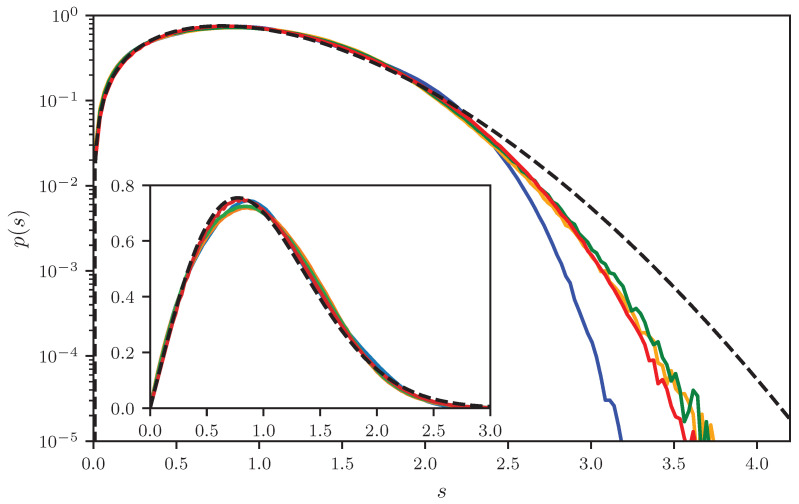
Level spacing distributions for the graphs shown in Table 1 in a logarithmic scale, with the tetrahedron (blue), the fully connected graph with 5 vertices (orange), the octahedron (green), and the hexahedron (red). The plots were generated by superimposing the results from $52.2\cdot 10^{6}$, $14.6\cdot 10^{6}$, $6.7\cdot 10^{6}$, $35.0\cdot 10^{6}$ spacings. The inset shows the same data in a linear scale. In addition, the exact RMT distribution is plotted (dashed black).

**Figure 5 entropy-25-00109-f005:**
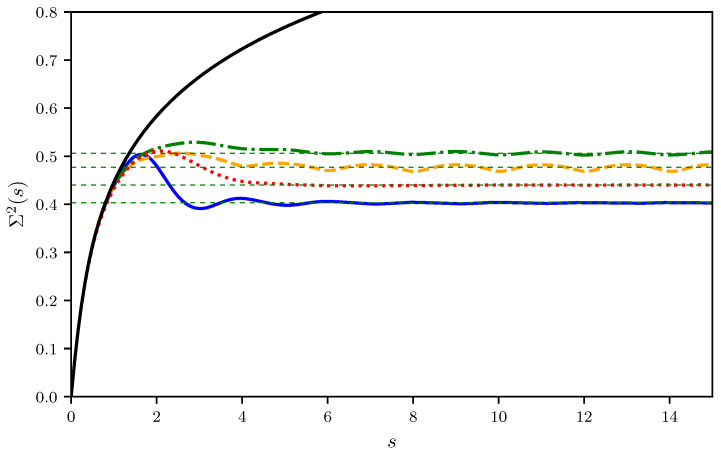
Number variance $\Sigma^{2}\left(s\right)$ for the tetrahedron (solid blue), the fully connected graph with 5 vertices (dashed orange), the octahedron (dashed dotted green), and the hexahedron (dotted red). The solid black line correspond to the RMT prediction, the horizontal thin dashed green lines mark the limiting values obtained from an average of $\Sigma^{2}\left(s\right)$ over the range s=10 to 20.

**Table 1 entropy-25-00109-t001:** The investigated graphs for Neumann boundary conditions at the vertices. The lower part of the table shows the results of the numerics: (i) $s_{c}$ is the *s* value, where p(s) drops below $10^{-5}$, i. e., $p\left(s_{c}\right)=10^{-5}$, (ii) $s_{m}$ is the *s* value, where $\Sigma^{2}\left(s\right)$ takes its maximal value, and (iii) $\Sigma_{\mathrm{sat}}^{2}$ is the limit of $\Sigma^{2}\left(s\right)$ for s→∞, obtained by taking the average of $\Sigma^{2}\left(s\right)$ in the range of *s* between 10 and 20.

Name	Tetrahedron	f. c. with V=5	Octahedron	Hexahedron
	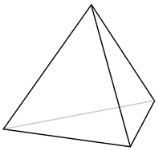	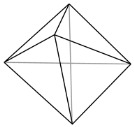	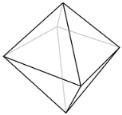	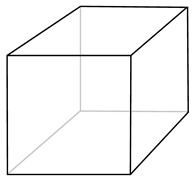
#Bonds *B*	6	10	12	12
#Vertices *V*	4	5	6	8
Valency of vertices	3	4	4	3
Fully connected (f. c.)	Yes	Yes	No	No
$s_{c}$	3.19	3.75	3.96	3.56
$s_{m}$	1.64	2.56	2.88	2.08
$\Sigma_{sat}^{2}$	0.40	0.48	0.51	0.44

## Data Availability

The data supporting the findings of this study are available from the corresponding authors upon request.

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
