# Peer review of "Implications of Spectral Interlacing for Quantum Graphs"

_entropy, 2023, doi:10.3390/e25010109_

Round 1

Reviewer 1 Report

The manuscript is well written and could be favorably considered for publication.   

However, the authors should explain in what way their results differ/add to  the ones obtained , in the paper : "On the level spacing distribution in quantum graphs" , by F. Barra and P. Gaspard,  Journal of Statistical Physics volume 101, pages283–319 (2000). Chapter 5.4:   Graphs with disconnected bonds.

Unfortunately, before this is done I could not recommend the manuscript for publication.

Author Response

- The manuscript is well written and could be favorably considered for publication.   

Thanks!

- However, the authors should explain in what way their results differ/add to  the ones obtained , in the paper : "On the level spacing distribution in quantum graphs" , by F. Barra and P. Gaspard,  Journal of Statistical Physics volume 101, pages283–319 (2000). Chapter 5.4:   Graphs with disconnected bonds.

We would like to thank the referee to pointing us to the paper of Barra and Gaspard, which we have now included in our manuscript. The main result for the Dirichlet graphs of this paper is Eq(49), which is the spacing distribution for a single graph. An ensemble averaging is not carried out at all, which is typically done on RMT and is the goal of our paper.
We have added a discussion on the Barra and Gaspard results after Eq.(22). 

Additionally, Barra and Gaspard also noticed already deviations from RMT of p(s) for small Neumann graphs. A corresponding remark is added in section 3.1

Reviewer 2 Report

Comments on “Implications of Spectral Interlacing for Quantum Graphs”

by Junjie Lu et al 

This paper gave an interesting observation showing that spectral interlacing phenomenon affects spectral statistics of quantum graphs when the number of bonds and vertices finite. The authors derived explicit formulas of short range and long range spectral statistics for the case of Dirichlet boundary condition and provided numerical results for the Neumann boundary condition. The paper is well-organized and clearly written and worth publishing in this special issue. The following is some suggestions to improve readability of the paper.  

- It would be helpful for non-experts to provide the definition of quantum graphs in the first place. The author started suddenly from the secular equation without any definition of the system. 

- What does “totally disintegrated graph” mean? 

- The author mentioned “the Neumann spectrum is described by random matrix theory only locally”. What does “locally” means?

- in page 10: The author mentioned “From the interlacing theorem one would expect a correlation with the bond number, ..”.  What is correlated with the bond numbers?

- The author gave a “qualitative” interpretation for the fact that s_{sat} for the Neumann spectrum is smaller than that predicted from periodic orbit theory. However, why the violation of the assumption that “all orbit lengths are uncorrelated” shorten s_{sat}?

Author Response

- This paper gave an interesting observation showing that spectral interlacing phenomenon affects spectral statistics of quantum graphs when the number of bonds and vertices finite. The authors derived explicit formulas of short range and long range spectral statistics for the case of Dirichlet boundary condition and provided numerical results for the Neumann boundary condition. The paper is well-organized and clearly written and worth publishing in this special issue. 

Thanks! 

- It would be helpful for non-experts to provide the definition of quantum graphs in the first place. The author started suddenly from the secular equation without any definition of the system. 

We have added a short introduction at the beginning of our manuscript, pointing out the relevant papers.

- What does “totally disintegrated graph” mean? 

``disintegrated'' means that the graph is dispatched/disassembled, meaning that at vertices the connection to the other bonds do not matter as due to the Dirichlet boundary condition there is no coupling anyway, i.e. the spectrum of the graph is defined purely by the number of the bonds and their lengths.
We have replaced ``disintegrated'', by ''disassembled',  perhaps a better word.

 - The author mentioned “the Neumann spectrum is described by random matrix theory only locally”. What does “locally” means?

The meaning becomes clear later: Because of the interlacing theorem there are at most V (number of vertices)  Neumann eigenvalues confined between two Dirichlet eigenvalues, meaning the RMT can work at most over a range of length V  (in units of the mean level spacing),  in accordance with the experimental and numerical findings. We changed the sentence in the abstract into  "... Neumann spectral statistics deviates from random matrix predictions."

- in page 10: The author mentioned “From the interlacing theorem one would expect a correlation with the bond number, ..”.  What is correlated with the bond numbers?

The remark refers to the quantities mentioned in the previous sentence, s_c, s_m, and Sigma^2_sat. We added an "of these quantities" for clarification.

- The author gave a “qualitative” interpretation for the fact that s_{sat} for the Neumann spectrum is smaller than that predicted from periodic orbit theory. However, why the violation of the assumption that “all orbit lengths are uncorrelated” shorten s_{sat}?

We have added a sentence at the end of paragraph 3 to detail when RMT is applicable. 
It was not stated that the qualitative interpretation explains the early saturation, it was only stated that the assumption of uncorrelated orbits is severely violated in graphs, and that therefore deviations from RMT predictions must not surprise. For good reasons nothing was said in detail about the consequences of this violation for the spectral statistics: We do not know!

Round 2

Reviewer 1 Report

I am satisfied with the corrections.

Reviewer 2 Report

The authors have responded to all the questioned I addressed. Hence, the paper should be published as it is.